# LEARNING FLEXIBLE CLASSIFIERS WITH SHOT-CONDITIONAL EPISODIC (SCONE) TRAINING

## ABSTRACT

Early few-shot classification work advocates for episodic training, i.e. training over learning episodes each posing a few-shot classification task. However, the role of this training regime remains poorly understood, and its usefulness is still debated. Standard classification training methods ("pre-training") followed by episodic fine-tuning have recently achieved strong results. This work aims to understand the role of this episodic fine-tuning phase through an exploration of the effect of the "shot" setting (number of examples per class) that is used during fine-tuning. We discover that fine-tuning on episodes of a particular shot can specialize the pre-trained model to solving episodes of that shot at the expense of performance on other shots, in agreement with a trade-off recently observed in the context of end-to-end episodic training. To amend this, we propose a shot-conditional form of episodic fine-tuning, inspired from recent work that trains a single model on a distribution of losses. Our investigation shows that this improves overall performance, without suffering disproportionately on any shot. We also examine the usefulness of this approach on the large-scale Meta-Dataset benchmark where test episodes exhibit varying shots and imbalanced classes. We find that our flexible model improves performance in that challenging environment.

## 1 INTRODUCTION

Few-shot classification is the problem of learning a classifier using only a few examples. Specifically, the aim is to utilize a training dataset towards obtaining a flexible model that has the ability to 'quickly' learn about new classes from few examples. Success is evaluated on a number of *test episodes*, each posing a classification task between previously-unseen *test* classes. In each such episode, we are given a few examples, or "shots", of each new class that can be used to adapt this model to the task at hand, and the objective is to correctly classify a held-out set of examples of the new classes.

A simple approach to this problem is to learn a classifier over the training classes, parameterized as a neural network feature extractor followed by a classification layer. While the classification layer is not useful at test time due to the class shift, the embedding weights that are learned during this "pre-training" phase evidently constitute a strong representation that can be used to tackle test tasks when paired with a simple "inference algorithm" (e.g. nearest-neighbour, logistic regression) to make predictions for each example in the test episode given the episode's small training set. Alternatively, early influential works on few-shot classification (Vinyals et al., 2016) advocate for *episodic training*, a regime where the training objective is expressed in terms of performance on a number of *training episodes* of the same structure as the test episodes, but with the classes sampled from the training set. It was hypothesized that this episodic approach captures a more appropriate inductive bias for the problem of few-shot classification and would thus lead to better generalization.

However, there is an ongoing debate about whether episodic training is in fact required for obtaining the best few-shot classification performance. Notably, recent work (Chen et al., 2019; Dhillon et al., 2020) proposed strong "pre-training" baselines that leverage common best practices for supervised training (e.g. normalization schemes, data augmentation) to obtain a powerful representation that works well for this task. Interestingly, other recent work combines the pre-training of a single classifier with episodic fine-tuning by removing the classification head and continuing to train the embedding network using the episodic inference algorithm that will be applied at test time (Triantafillou et al., 2020; Chen et al., 2020). The success of this hybrid approach suggests that perhaps the two regimes

have complementary strengths, but the role of this episodic fine-tuning is poorly understood: what is the nature of the modification it induces into the pre-trained solution? Under which conditions is it required in order to achieve the best performance?

As a step towards answering those questions, we investigate the effect of the shot used during episodic fine-tuning on the resulting model's performance on test tasks of a range of shots. We are particularly interested in understanding whether the shot of the training episodes constitutes a source of information that the model can leverage to improve its few-shot classification performance on episodes of that shot at test time. Our analysis reveals that indeed a particular functionality that this fine-tuning phase may serve is to specialize a pre-trained model to solving tasks of a particular shot; accomplished by performing the fine-tuning on episodes of that shot. However, perhaps unsurprisingly, we find that specializing to a given shot comes at the expense of hurting performance for other shots, in agreement with (Cao et al., 2020)'s theoretical finding in the context of Prototypical Networks (Snell et al., 2017) where inferior performance was reported when the shot at training time did not match the shot at test time.

Given those trade-offs, how can our newfound understanding of episodic fine-tuning as shot-specialization help us in practice? It is unrealistic to assume that we will always have the same number of labeled examples for every new class we hope to learn at test time, so we are interested in approaches that operate well on tasks of a range of shots. However, it is impractical to fine-tune a separate episodic model for every shot, and intuitively that seems wasteful as we expect that tasks of similar shots should require similar models. Motivated by this, we propose to train a single shot-conditional model for specializing the pre-trained solution to a wide spectrum of shots without suffering trade-offs. This leads to a compact but flexible model that can be conditioned to be made appropriate for the shot appearing in each test episode.

In what follows we provide some background on few-shot classification and episodic models and then introduce our proposed shot-conditioning approach and related work. We then present our experimental analysis on the effect of the shot chosen for episodic fine-tuning, and we observe that our shot-conditional training approach is beneficial for obtaining a general flexible model that does not suffer the trade-offs inherent in naively specializing to any particular shot. Finally, we experiment with our proposed shot-conditional approach in the large-scale Meta-Dataset benchmark for few-shot classification, and demonstrate its effectiveness in that challenging environment.

## 2 BACKGROUND

**Problem definition** Few-shot classification aims to classify test examples of unseen classes from a small labeled training set. The standard evaluation procedure involves sampling *classification episodes* by picking $N$ classes at random from a test set of classes $\mathcal{C}^{test}$ and sampling two disjoint sets of examples from the $N$ chosen classes: a *support* set (or training set) of $k$ labeled examples per class, and a *query* set (or test set) of unlabeled examples, forming $N$-way, $k$-shot episodes. The model is allowed to use the support set, in addition to knowledge acquired while training on a disjoint set of classes $\mathcal{C}^{train}$, to make a prediction for examples in the query set, and is evaluated on its query set accuracy averaged over multiple test episodes.

**Episodic training** Early few-shot classification approaches (Vinyals et al., 2016) operate under the assumption that obtaining a model capable of few-shot classification requires training it on (mini-batches of) learning episodes, instead of (mini-batches of) individual examples as in standard supervised learning. These learning episodes are sampled in the same way as described above for test episodes, but with classes sampled from $\mathcal{C}^{train}$ this time. In other words, the model is trained to minimize a loss of the form:

$$\mathbb{E}_{\mathcal{S}, \mathcal{Q} \sim P_{train}^{N,k}} \left[ \frac{1}{|\mathcal{Q}|} \sum_{(x^*, y^*) \in \mathcal{Q}} -\log p_\theta(y^* \mid x^*, \mathcal{S}) \right] \tag{1}$$

where $\mathcal{S}$ and $\mathcal{Q}$ are support and query sets sampled from the distribution $P_{train}^{N,k}$ of $N$-way, $k$-shot training episodes induced by $\mathcal{C}^{train}$, and $\theta$ represents the model's parameters. This training regime is often characterized as *meta-learning* or *learning to learn*, i.e. learning over many episodes how to learn within an episode (from few labeled examples). Episodic models differ by their "inference

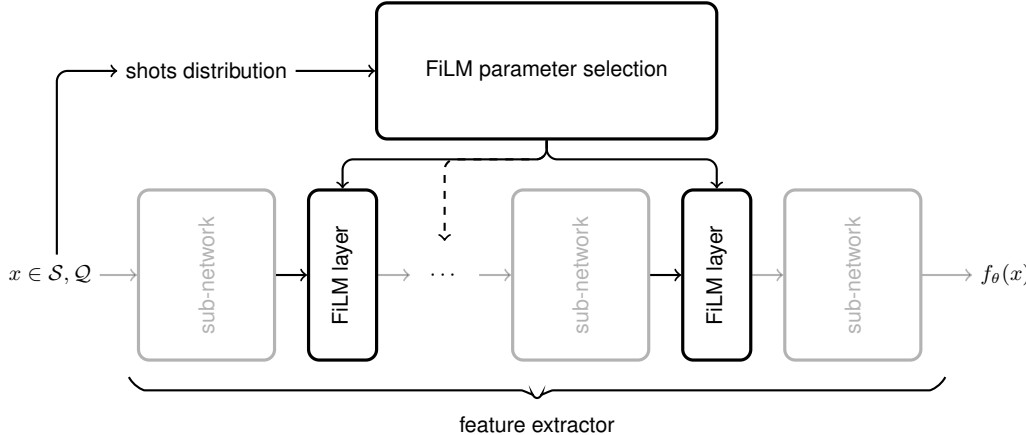

Figure 1: SCONE conditions the feature extractor $f_\theta$ on an episode's shot distribution.

algorithm", i.e. the manner in which $p_\theta(y^* \mid x^*, \mathcal{S})$ is computed to classify query examples based on the support set.

**Prototypical Networks**   Prototypical Networks (Snell et al., 2017) is a simple but effective episodic model which constructs a prototype $\phi_c$ for each class $c$ in an episode as

$$\phi_c = \frac{1}{|\mathcal{S}_c|} \sum_{x \in \mathcal{S}_c} f_\theta(x), \tag{2}$$

where $f$ is an embedding function parametrized by $\theta$ and $\mathcal{S}_c$ represents the set of support examples belonging to class $c$, and classifies a given query example as

$$p(y^* = c \mid x^*, \mathcal{S}) = \frac{\exp(-||x^* - \phi_c||_2^2)}{\sum_{c'} \exp(-||x^* - \phi_{c'}||_2^2)}. \tag{3}$$

## 3   SHOT CONDITIONAL EPISODIC (SCONE ) TRAINING

In this section we introduce Shot CONditional Episodic (SCONE ) training for the purpose of specializing a strong pre-trained model to solving few-shot classification tasks of a range of different shots, without suffering disproportionately for any shot.

**Training objective**   Training episodically involves minimizing the objective shown in Equation 1. We first sample an episode from $P_{train}^{k,N}$ and compute a prediction $p_\theta(y^* \mid x^*, \mathcal{S})$ for each query example $x^*$. We then compute the cross-entropy loss on the query set using those predictions and perform a parameter update by backpropagating its gradient with respect to $\theta$ *into the inference algorithm*. In this work we concern ourselves with models that use an embedding function $f_\theta$ to obtain a representation for the support and query examples of each episode on top of which the inference algorithm is applied. In Prototypical Networks, for instance, $f_\theta$ contains *all* of the model's learnable parameters.

SCONE trains on episodes of varying shots and conditions the model on each episode's shot distribution. (Figure 1) by minimizing

$$\mathbb{E}_{k \sim P_k} \left[ \mathbb{E}_{\mathcal{S}, \mathcal{Q} \sim P_{train}^{N,k}} \left[ \frac{1}{|\mathcal{Q}|} \sum_{(x^*, y^*) \in \mathcal{Q}} - \log p_{\theta_k}(y^* \mid x^*, \mathcal{S}) \right] \right], \tag{4}$$

where $P_k$ is the distribution over shots at training time and $\theta_k$ depends on an episode's sampled shots. In the Appendix, we include an algorithm box outlining SCONE fine-tuning.

**Conditioning mechanism** Rather than learning a separate set of model parameters for each shot setting, we modulate a subset of its parameters using FiLM (Perez et al., 2018), a simple conditioning mechanism which performs an affine feature-wise transformation of its input $x$ based on conditioning information $k$ (in our case, the episode's number of shots):

$$\text{FiLM}(x) = \gamma(k) \odot x + \beta(k). \tag{5}$$

The dependency of $\gamma$ and $\beta$ on $k$ is handled by maintaining distinct values for each shot setting and selecting the appropriate $\gamma$ and $\beta$ based on an episode's shot. Equivalently, we can think of our approach as a compact representation of many shot-specific feature extractors which share all but their FiLM layer parameters.

More concretely, we maintain a set of FiLM parameters for each shot in the [1, MAX-SHOT] range (where MAX-SHOT is a hyperparameter) and let all shots settings greater than or equal to MAX-SHOT share the same FiLM parametrization. As is often the case in practice, instead of inserting FiLM layers in the network's architecture, we modulate the scaling and shifting parameter values of existing batch normalization layers (Dumoulin et al., 2017; De Vries et al., 2017). When performing episodic fine-tuning, we initialize all sets of FiLM parameters to those learned during pre-training (i.e. the learned batch normalization scaling and shifting coefficients). These different sets of FiLM parameters are then free to deviate from each other as a result of fine-tuning. We found it beneficial to penalize the L2-norm of $\beta$ (regularizing the offset towards 0) and the L2 norm of $\gamma - 1$ (regularizing the scaling towards 1). For this purpose, we introduce a hyperparameter that controls the strength of this FiLM weight decay.

**Handling class-imbalanced episodes** SCONE can also be used on imbalanced episodes, where different classes have different shots. In that case, instead of selecting a single set of FiLM parameters, we compute the FiLM parameters for an episode as the convex combination of the FiLM parameters associated with all shots found in the episode, where the weights of that combination are determined based on the frequency with which each shot appears in the episode.

Concretely, the episode's "shot distribution" $s$ (a vector of length MAX-SHOT) is obtained by averaging the one-hot representations of the shots of the classes appearing in an episode. In the special case of a class-balanced episode, the resulting average will be exactly a one-hot vector. This shot distribution is then used for the purpose of selecting the episode's FiLM parameters. This can be thought of as an embedding lookup $s^T \mathcal{F}$ in a matrix $\mathcal{F}$ of FiLM parameters using a shot distribution $s$.

**Smoothing the shot distribution** We expect similar shot values to require similar FiLM parameters, which we incorporate as an inductive bias by smoothing the shot distribution. We outline our SMOOTH-SHOT procedure in the Appendix in Algorithm 1, which receives the shot $s$ of a class (an integer), and a smoothing hyperparameter $m$ (a float in [0, 1]) and returns the smoothed shot for that class, which is a vector of length MAX-SHOT. Essentially, the result of smoothing is that the returned vector representation of $s$ is not strictly one-hot with only the position corresponding to the observed shot $s$ being 'on'. Instead, some entries surrounding that position are also non-zero. Specifically, the entries that are directly adjacent to $s$ receive the value $m$, the entries two spots away from $s$ the value $m^2$, and so on, with entries further away from $s$ receiving exponentially-decaying values.

## 4 RELATED WORK

**Few-shot classification** A plethora of models have been recently proposed for few-shot classification, and we refer the reader to (Hospedales et al., 2020) for a broad survey. Before episodic training was introduced, few-shot classifiers often relied on metric learning (Koch et al., 2015; Triantafillou et al., 2017). This theme persisted in early episodic models like Matching Networks (Vinyals et al., 2016) and Prototypical Networks (Snell et al., 2017) where classification is made via nearest-neighbour comparisons in the embedding space. Matching Networks apply a soft $k$-NN algorithm where the label of a query example is predicted to be the weighted average of the (one-hot) support labels with the weights determined by the similarity of that query to each support example.

Gradient-based episodic models are another popular family of approaches following the influential MAML paper (Finn et al., 2017). To create a classifier for each given episode, this approach fine-tunes the embedding weights along with a linear classifier head using gradient descent on the support set.

Intuitively, this results in learning an embedding space that serves as a useful starting point from which a few steps of gradient descent suffice to adapt the model to each episode's classification task. Proto-MAML (Triantafillou et al., 2020) is a simple extension that initializes the linear classifier for each episode from the prototypes of the classes appearing in that episode.

Recently, the field has shifted towards studying few-shot classification in more realistic environments like *tiered*-ImageNet (Ren et al., 2018) and Meta-Dataset (Triantafillou et al., 2020), which has encouraged research into newly-introduced challenges, such as accounting for multiple diverse datasets. Along these lines, Requeima et al. (2019); Bateni et al. (2019) proposed novel task conditioning approaches, Saikia et al. (2020) introduced an improved hyperparameter tuning approach, and Dvornik et al. (2020) proposed a method for selecting an appropriate set of features for each test episode out of a universal feature representation.

**Understanding episodic learning**  Our work inscribes itself in a recent line of work attempting to understand the differences between episodic and non-episodic learning. Goldblum et al. (2020) attempts to understand episodic learning from the perspective of how classes cluster in feature-space (for models that learn a final classification layer on top of a feature extractor) as well as from the perspective of local minima clusters (for gradient-based meta-learners). Huang et al. (2020); Chao et al. (2020) draw parallels between learning episodes and supervised learning examples, Bronskill et al. (2020) discusses batch normalization in episodic learning, drawing parallels from its use in non-episodic learning and Chen et al. (2020) contrasts episodic and non-episodic learning in their ability to generalize to new examples of previously seen classes or new examples of *unseen* classes. Finally, Cao et al. (2020) theoretically investigates the role of the shot in Prototypical Networks to explain the observed performance drop when there is a mismatch between the shots at training and test time. Instead, we empirically study the effect of the shot chosen during episodic fine-tuning of a pre-trained solution, in a larger-scale and more diverse environment.

**Feature-wise conditioning**  Feature-wise transformations such as FiLM (Perez et al., 2018) are used as a conditioning mechanism in a variety of problem settings; see Dumoulin et al. (2018) for a survey on the topic. (Shu et al., 2019) devise a loss re-weighting scheme that conditions on the loss at each time-step, which is a scalar, thus bearing similarity to our approach when conditioning on a scalar shot setting. In few-shot classification, (Sun et al., 2019) use feature-wise transformations as a means of transfer to new tasks. (Oreshkin et al., 2018; Requeima et al., 2019; Bateni et al., 2019) use FiLM to condition metric learners' backbones on the support set, while (Dvornik et al., 2020) uses it as a way to represent many pre-trained classifiers using a shared parametrization. FiLM has also been used successfully for class-incremental learning Liu et al. (2020) and semi-supervised few-shot learning (Li et al., 2019). Notably, TADAM (Oreshkin et al., 2018), CNAPs (Requeima et al., 2019) and Simple-CNAPs (Bateni et al., 2019) also use task conditioning, but they use the mean of the support set for this and thus the 'shot' information is discarded. The purpose of our conditioning mechanism is instead to make the backbone shot-aware. The idea of shot-conditional learners is inspired by recent work that investigates loss-conditional training using feature-wise transformations (Dosovitskiy & Djolonga, 2020; Babaeizadeh & Ghiasi, 2020).

## 5 EXPERIMENTS

### 5.1 EXPLORING THE ROLE OF 'SHOTS' DURING EPISODIC FINE-TUNING

In this subsection, we examine the effect of the 'shot' that is used during the episodic fine-tuning phase, and in particular how it impacts the resulting model's ability to solve test episodes of different shots. We consider either using a fixed shot $k$ throughout the fine-tuning phase, or fine-tuning on episodes of a distribution of shots. In the latter case, we explore both standard fine-tuning as well as SCONE fine-tuning that equips the model with the shot-conditioning mechanism described in the previous section. We also compare against EST (Cao et al., 2020).

**Experimental setup**  We ran this round of experiments on ImageNet using the class splits proposed in Meta-Dataset. First, we pre-trained a standard classifier on the set of training classes of ImageNet. We then removed the topmost classification layer, leaving us with a pre-trained backbone that we used as the initialization for the subsequent episodic fine-tuning round. We ran the following

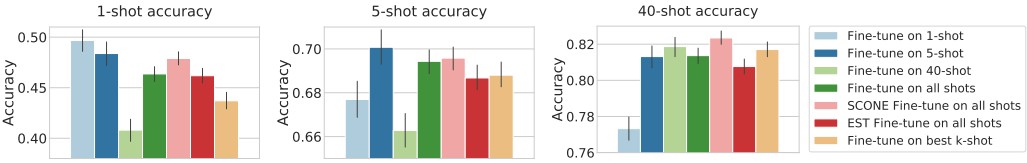

Figure 2: Test accuracy on three different evaluation shots. Fine-tuning exclusively on a particular shot leads to the best test accuracy on that shot but poor accuracy on different shots. Fine-tuning on a range of shots is a reasonable general solution, but its performance can be improved when using SCONE , thanks to its conditioning mechanism that offers a compact form of shot specialization.

variants of episodic fine-tuning: exclusively on 1-shot episodes ('Fine-tune on 1-shot'), exclusively on 5-shot episodes ('Fine-tune on 5-shot'), on episodes whose shot is drawn uniformly from the range $[1, 40]$ ('Fine-tune on all shots'), and on episodes with that same shot distribution but using SCONE ('SCONE Fine-tune on all shots'), which additionally equips the backbone with the shot conditioning mechanism described in the previous section. We also consider 'Fine-tune on best k-shot', an additional baseline that fine-tunes exclusively on the shot $k$ that is found to work best on average on the validation set (on the range of shots $1 - 40$). For this, we trained models for $k = 1, 5, 10, 15, 20, 30, 40$ and found the best to be $k = 15$.

As mentioned in the previous section, when applying SCONE training, we penalize the L2 norm of FiLM parameters. For a fair comparison with the other models, we applied the same regularization to the batch normalization parameters of all models during the episodic fine-tuning phase, and we found this to be generally helpful. We tuned the strength of this regularization separately for each model and picked the variant that worked best on the validation set, which we report in the Appendix. We set the SCONE 's MAX-SHOT hyperparameter to be 40 for this experiment.

We also compare to EST (Cao et al., 2020) which is a theoretically-grounded method for building shot resiliency in Gaussian classifiers. This involves applying a linear transformation on top of the learned embeddings that aims to strike a good balance between maximizing the inter-class variance and minimizing the intra-class variance. In practice, that trade-off is controlled via a hyperparameter $\rho$. We applied EST on top of the embeddings of 'Fine-tune on all shots' and we tuned $\rho$ and the hyperparameter $d$ controlling the projection dimensionality very extensively. The values that we found worked best (selected on the validation set of ImageNet on the range of shots 1-40) are substantially different than those used in the original EST paper: $d = 480$ and $\rho = 5e - 8$ (versus the original $d = 120$ and $\rho = 1e - 3$). We believe that this discrepancy may be due to our deeper backbones and larger range of shots. The EST configuration that worked best for us yields a minimal reduction in the embedding dimensionality, and primarily favours maximizing the inter-class variance, with the term that minimizes the intra-class variance having minimal effect.

In all cases, we fix the 'way' to 5. We use Prototypical Networks as the episodic model and we perform early stopping and model selection on the validation set of classes, where the validation performance of a variant is computed on episodes of the same (distribution of) shot(s) that it is trained on. All models are tested on a held-out test set of classes that is not seen during pre-training nor episodic fine-tuning, on 5-way episodes of different shot settings.

**Findings**    We observe from Figure 2 that fine-tuning on a fixed shot yields the best results on test episodes of that shot. For example, 1-shot accuracies show that 'Fine-tune on 1-shot' surpasses the performance of all other variants on 1-shot test episodes, with the analogous findings in 1-shot and 5-shot accuracies for 5-shot and 40-shot, respectively. Therefore, a particular functionality that the episodic fine-tuning phase may serve is to specialize the pre-trained model for performing well on tasks of a particular shot. However, as illustrated in all sub-plots of Figure 2, this shot specialization comes at the cost of severely reduced performance on tasks of very different shots. For instance, the model that is specialized for 40-shot tasks ('Fine-tune on 40-shot') performs very poorly on 1-shot test tasks and vice-versa. We also note that the 'Fine-tune on best k-shot' model does not suffice to perform well in all settings either, since $k = 15$ there, it performs really poorly on 1-shot for instance.

In practice, it may be desirable to perform well on more than a single shot setting at test time, without having to fine-tune multiple separate shot-specialized models. A reasonable approach to that is

| Dataset | Prototypical Networks (ImageNet only) | | | | | |
|---|---|---|---|---|---|---|
| | Standard | L2 BN | EST | Best k-shot | SCONE w/o S | SCONE |
| ILSVRC-2012 | 50.90 ± 1.12% | **51.81 ± 1.06%** | 52.17 ± 1.09% | 52.36 ± 1.08% | **52.98% ± 1.09%** | 52.51 ± 1.11% |
| Omniglot | 63.12 ± 1.37% | 63.14 ± 1.32% | **66.07 ± 1.29%** | 65.94 ± 1.33% | 64.71% ± 1.32% | 65.60 ± 1.34% |
| Aircraft | 54.30 ± 0.97% | 53.26 ± 0.97% | **55.64 ± 0.88%** | 56.03 ± 0.95% | 55.38% ± 0.96% | 55.38 ± 0.96% |
| Birds | 68.22 ± 0.97% | **69.21 ± 1.01%** | 67.17 ± 1.02% | 68.63 ± 1.11% | 68.98% ± 1.04% | 69.70 ± 1.01% |
| DTD | 66.62 ± 0.90% | 68.33 ± 0.81% | 68.20 ± 0.77% | 69.61 ± 0.80% | 68.68% ± 0.78% | 69.58 ± 0.77% |
| Quickdraw | **59.79 ± 0.98%** | 59.17 ± 0.96% | 60.05 ± 0.97% | 60.68 ± 0.97% | 60.00% ± 1.00% | 60.81 ± 0.95% |
| Fungi | 36.77 ± 1.07% | **38.96 ± 1.10%** | 39.50 ± 1.12% | 37.96 ± 1.08% | 39.19% ± 1.15% | 39.66 ± 1.12% |
| VGG Flower | 86.61 ± 0.87% | **87.70 ± 0.77%** | 88.55 ± 0.65% | 87.45 ± 0.86% | 86.98% ± 0.80% | 88.03 ± 0.73% |
| Traffic Signs | 48.64 ± 1.06% | 46.54 ± 1.03% | 48.41 ± 1.07% | **50.26 ± 1.16%** | 47.61% ± 1.05% | 48.24 ± 1.09% |
| MSCOCO | **43.02 ± 1.09%** | 43.11 ± 1.05% | 43.45 ± 1.05% | 43.20 ± 1.18% | 43.43% ± 1.08% | 44.25 ± 1.11% |
| Average | 57.80 % | 58.12 % | 58.92% | 59.21% | 58.79% | 59.38% |
| Average ranks | 4.90 | 4.10 | 3.25 | 3.05 | 3.00 | **2.70** |

Table 1: Prototypical Networks fine-tuned on ImageNet ('Standard') with the addition of L2 regularization on the batch normalization weights ('L2 BN'), EST (Cao et al., 2020), the 'Fine-tune on best k-shot' baseline ('Best $k$-shot') and SCONE , including the ablation that omits the shot smoothing ('SCONE w/o S'). The reported numbers are query set accuracies averaged over 600 test episodes and 95% confidence intervals. We also show the average ranks (lower is better). We report details in the Appendix on rank computation and statistical testing.

episodically fine-tuning on a range of shots, to obtain a general model. Indeed, Figure 2 shows that 'Fine-tune on all shots' does not perform too poorly on any shot but, perhaps unsurprisingly, in any given setting, it falls short of the performance of the corresponding shot-specialized model.

Finally, we observe that SCONE fine-tuning outperforms its shot-unaware counterpart in all settings ('SCONE Fine-tune on all shots' vs 'Fine-tune on all shots'). This constitutes evidence that SCONE fine-tuning indeed leads to a more flexible model that can adapt to the shot of each episode via its conditioning mechanism, without suffering the trade-offs inherent in naively specializing a model exclusively to any particular shot. We can view a SCONE model as a very compact way of representing multiple shot-specialized models, where the information required for that specialization resides in the light-weight FiLM parameters. SCONE also outperforms the EST approach in this setting which also strives for shot resiliency, but does so by encouraging invariance to the shot setting rather than shot awareness as in SCONE .

## 5.2 LARGE-SCALE EXPERIMENTS ON META-DATASET

In what follows, we apply SCONE to the diverse and challenging Meta-Dataset benchmark for few-shot classification (Triantafillou et al., 2020). Meta-Dataset is comprised of ten distinct image datasets, including natural images, handwritten characters and sketches. It also defines a generative process for episodes that varies the way and shot across episodes, and within a particular episode varies the shot for different classes, introducing imbalance. The range of shots induced by this episode generator is also larger than what we considered in the previous section. It is a long-tailed distribution under which small and medium shots are more likely but it is possible to also encounter very large shots (e.g. >400), though this would happen very infrequently. We include histograms of the shot distributions of Meta-Dataset's training, validation and test episodes in the Appendix. These experiments aim to investigate whether SCONE is effective on this broader shot distribution and imbalanced episodes.

**Prototypical Network on ImageNet** For our first set of experiments on Meta-Dataset, we explore different strategies of episodic fine-tuning of the pre-trained classifier's embedding weights using Prototypical Networks. For this, we use Meta-Dataset's sampling algorithm to draw training episodes of varying shots and ways from ImageNet. We compare standard episodic fine-tuning ('Standard') to SCONE episodic fine-tuning ('SCONE '). Since SCONE uses L2-regularization on the sets of FiLM parameters, for a fair comparison we include a variant of standard episodic fine-tuning with L2-regularization on the batch normalization parameters ('L2 BN'). We also include an ablation of our method that does not use any smoothing of the shot distribution. Finally, we compare to EST as well where we computed the EST transformation on the 'L2 BN' instead of the 'Standard' Prototypical Network variant, since that worked best. We tuned EST's hyperparameters very extensively, as

| Dataset | Meta-Baseline (All datasets) | | |
|---|---|---|---|
| | Classifier-Baseline | Control | SCONE |
| ILSVRC-2012 | **53.44 ± 0.82%** | 49.83 ± 0.80% | **53.69 ± 0.83%** |
| Omniglot | 81.66 ± 0.73% | **89.28 ± 0.51%** | **90.01 ± 0.49%** |
| Aircraft | 70.65 ± 0.62% | **81.60 ± 0.49%** | 78.27 ± 0.54% |
| Birds | 76.99 ± 0.64% | **78.75 ± 0.59%** | **79.62 ± 0.58%** |
| DTD | **71.28 ± 0.56%** | 70.47 ± 0.58% | **71.89 ± 0.59%** |
| Quickdraw | 64.09 ± 0.67% | **72.79 ± 0.59%** | 71.95 ± 0.56% |
| Fungi | 50.23 ± 0.81% | 55.28 ± 0.73% | **57.04 ± 0.74%** |
| VGG Flower | 89.14 ± 0.44% | 90.13 ± 0.43% | **91.09 ± 0.39%** |
| Traffic Signs | 89.14 ± 0.44% | 90.13 ± 0.43% | **91.09 ± 0.39%** |
| MSCOCO | **53.92 ± 0.78%** | 47.85 ± 0.81% | 52.94 ± 0.82% |
| Average | 68.03% | 70.63% | 71.68% |
| Average ranks | 2.55 | 2 | **1.45** |

Table 2: Our reproduction of the Classifier-Baseline (Chen et al., 2020) trained on all datasets, and two variants that freeze those weights and episodically fine-tune using Meta-Baseline (Chen et al., 2020) to optimize either only the batch norm parameters ('Control'), or only SCONE 's parameters ('SCONE '). In all cases, the reported numbers are query set accuracies averaged over 1K test episodes and 95% confidence intervals. We also show the average ranks (lower is better).

described in Section 5.1, this time model-selecting on the validation sets of all datasets of Meta-Dataset. The values that worked best in this case are $d = 480$ and $\rho = 5e - 9$. As noted in Section 5.1, these are substantially different than those used in the original EST paper, likely due to our deeper backbones and significantly broader range of shots explored. Finally, we ran the same 'Fine-tune on best k-shot' baseline described in Section 5.1. In this case we found that the best $k$ was 20.

We evaluate these models on the held-out set of ImageNet classes as well as the remaining 9 datasets. We set SCONE 's MAX-SHOT to 200. We tune the learning rate and decay schedule separately for each variant and we perform model selection of SCONE 's hyperparameters using the validation set. All additional details are reported in the Appendix, and we plan to open source our code upon publication.

**Meta-Baseline on all datasets** Next, we experiment with the recent Meta-Baseline model (Chen et al., 2020). Meta-Baseline also consists of a pre-training phase ('Classifier-Baseline') followed by an episodic fine-tuning phase ('Meta-Baseline'). Classifier-Baseline refers to simply training a classifier on the set of training classes. This variant is evaluated on few-shot episodes by discarding the ultimate classification layer and utilizing a cosine similarity-based nearest-centroid inference algorithm on the learned embeddings. Meta-Baseline then fine-tunes Classifier-Baseline's pre-trained embeddings on the episodic objective of the aforementioned nearest-centroid algorithm.

When training on all datasets of Meta-Dataset, they obtained strong results using their Classifier-Baseline which is in this case trained in a multi-task setup with separate output heads for the different datasets. They found that episodically fine-tuning that solution on all datasets did not help in general (it improved performance on some datasets but hurt performance on a larger number of datasets).

Inspired by that finding, we experimented with a SCONE training phase on top of Classifier-Baseline's strong pre-trained solution where we froze the embedding weights to that powerful representation and we optimized only the set of SCONE 's FiLM parameters for shot conditioning. We performed this fine-tuning on training episodes from all datasets, using Meta-Baseline's nearest centroid method as the episodic model. As a control experiment, we performed the same episodic fine-tuning but without shot-conditioning, where we optimized only the batch normalization parameters, keeping the remainder of the

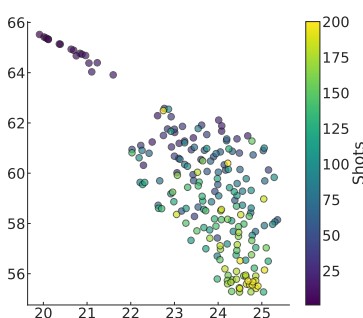

Figure 3: UMAP projection of the learned FiLM parameters for each "shot" setting, color-coded by shots.

embedding weights frozen ('Control'). This control can be thought of as a special case of SCONE where MAX-SHOT is set to 1.

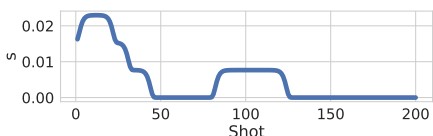

Figure 4: The shot distribution $s$ produced according to our smoothing procedure for a hypothetical 4-way episode where the shots for the four classes are: 1, 10, 23, and 103.

**Findings**    The results of this investigation are shown in Table 1 and Table 2 (as well as their more heavily-annotated counterparts in the Appendix, Tables 4 and 5, that show the per-row rank computation). Following (Triantafillou et al., 2020), we run a test of statistical significance described in the Appendix to determine when to bold an entry. Table 1 shows that SCONE fine-tuning outperforms standard episodic fine-tuning in the context of Prototypical Networks. Interestingly, penalizing the L2-norm of batch normalization parameters during episodic fine-tuning is beneficial even when not using SCONE , but it does not reach the performance obtained by our shot-conditioning. The ablation of SCONE that does not use any smoothing of the shot distribution is also competitive, but performs worse than full SCONE . We also observe that EST is competitive in this setting, only slightly worse than SCONE , though we note that SCONE is a more general approach that is not tied to Gaussian classifiers. Similarly, in the context of Meta-Baseline, Table 2 shows that episodically fine-tuning the batch normalization parameters of the otherwise-frozen embedding is helpful ('Control'), but using SCONE to learn a separate set of FiLM parameters for each shot yields additional gains in this setting too. Overall, despite the simplicity of SCONE , these results demonstrate its effectiveness on different shot distributions, and in different backbones.

**FiLM parameter visualization**    Finally, as a sanity check, we perform a UMAP projection (McInnes et al., 2018) of the learned FiLM parameters for each shot setting (Figure 3). As expected, similar shot settings tend to learn similar sets of FiLM parameters, which is reflective of the fact that they rely on similar features for classification.

**Example smoothed shot distribution**    To gain an intuition on the effect of our smoothing procedure, we illustrate in Figure 4 the result of smoothing an example shot distribution using $m = 1 - 1e - 06$, which is the value of the smoothing hyperparameter that we used for our Prototypical Network experiments on Meta-Dataset. For this, we consider a hypothetical 4-way episode where the shots for the four classes are: 1, 10, 23, and 103. We observe that the largest peak is in the range of small values, due to the first three shots of the episode, with the fourth shot causing a second peak around the value 103. As a reminder, this shot distribution defines the weights of the convex combination of FiLM parameters that will be used for the episode. In practice therefore, we are activating 'blocks' of FiLM parameters that are relevant for each episode, instead of strictly activating only the FiLM parameters of the observed shots.

## 6    CONCLUSION

In summary, we present an analysis aiming to understand the role of episodic fine-tuning on top of a pre-trained model for few-shot classification. We discover that this fine-tuning phase can be used to specialize the pre-trained model to episodes of a given shot, leading to strong performance on test episodes of that shot at the expense of inferior performance on other shots. To eliminate that trade-off, we propose a shot-conditional episodic training approach that trains a model on episodes of a range of shots and can be conditioned at test time to modify its behavior appropriately depending on the shot of the given test episode. Our experimental analysis suggests that our proposed shot-conditioning mechanism is beneficial both in smaller-scale experiments, as well as in the large-scale and diverse Meta-Dataset benchmark, in the context of two different episodic models. Future work could explore how to incorporate shot-awareness in other few-shot classification models. In addition to the architectural modification of FiLM conditioning on the shot distribution, are there algorithmic adjustments that can yield additional performance gains, such as a mechanism of determining the number of inner-loop updates to perform for gradient-based meta-learners based on the number of available shots?

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

## A  APPENDIX

### SCONE 's training algorithm in more detail

For clarity, we provide pseudocode for SCONE 's training algorithm including our procedure for shot smoothing in Algorithm 1. We will also release our code upon publication for reproducibility.

---

**Algorithm 1** SCONE training

---

**Input:** Distributions of training episodes $P_{train}$, pre-trained embedding weights $\theta$, pre-trained batch norm weights $\gamma$ and $\beta$, embedding function $f$, learning rate $\epsilon$ (a float), smoothing co-efficient $m$ (a float in the range $[0, 1]$) and maximum supported shot MAX-SHOT (an int).

**Output:** Finetuned embedding weights $\theta'$ and FiLM parameters $\mathcal{F} = \{\gamma', \beta'\}$.

 

    **procedure** SMOOTH-SHOT($s, m,$ MAX-SHOT)
        **if** $s >$ MAX-SHOT **then**
            $s \leftarrow$ MAX-SHOT                      $\triangleright$ Cap $s$ to the max supported shot
        **end if**
        $s \leftarrow s - 1$             $\triangleright$ So that $s$ is in the range $[0,$ MAX-SHOT $- 1]$
        $\tilde{s} \leftarrow$ ONE-HOT($s,$ DEPTH=MAX-SHOT)       $\triangleright$ Init the smoothed shot
        **for** $0 \leq j \leq$ MAX-SHOT **do**
            $l \leftarrow s - j - 1$          $\triangleright$ The index $j$ slots to the left of $s$
            $l \leftarrow$ ONE-HOT($l,$ DEPTH=MAX-SHOT) $* m$    $\triangleright$ Outputs the zero vector if $l < 0$
            $r \leftarrow s + j + 1$          $\triangleright$ The index $j$ slots to the right of $s$
            $r \leftarrow$ ONE-HOT($r,$ DEPTH=MAX-SHOT) $* m$    $\triangleright$ Outputs the zero vector if $r < 0$
            $\tilde{s} \leftarrow \tilde{s} + l + r$
            $m \leftarrow m^2$            $\triangleright$ Adjust the next iteration's smoothing
        **end for**
    **end procedure**

 

    $\theta' \leftarrow \theta$            $\triangleright$ Init the embedding weights from the pre-trained embeddings
    **for** $1 \leq k \leq$ MAX-SHOT **do**      $\triangleright$ Init the FiLM params from the pre-trained batch norm params
        $\gamma'(k) \leftarrow \gamma$
        $\beta'(k) \leftarrow \beta$
    **end for**
    **while** validation accuracy is improving **do**
        Sample a training episode with support set $\mathcal{S}$ and query set $\mathcal{Q}$
        Let $k_1, \ldots k_N$ be the shots of the episode's classes.
        $s \leftarrow$ ZEROS(MAX-SHOT)          $\triangleright$ Init the (unnormalized) shot distribution
        **for** each class $i$ **do**
            $s_i \leftarrow$ ONE-HOT($k_i,$ DEPTH $=$ MAX-SHOT)
            $s_i \leftarrow$ SMOOTH-SHOT($s_i, m,$ MAX-SHOT)      $\triangleright$ Smooth the one-hot shot of class $i$
            $s \leftarrow s + s_i$
        **end for**
        $s \leftarrow s \div$ SUM($s$)          $\triangleright$ Normalize to get the episode's shot distribution
        $\gamma'_s \leftarrow s^T \gamma'$          $\triangleright$ Select the FiLM params for the episode
        $\beta'_s \leftarrow s^T \beta'$
        Let $\mathcal{S}^H = \{f(x; \theta', \gamma'_s, \beta'_s), y\}_{(x,y) \in \mathcal{S}}$        $\triangleright$ The embedded support set
        Let $\mathcal{Q}^H = \{f(x; \theta', \gamma'_s, \beta'_s), y\}_{(x,y) \in \mathcal{Q}}$        $\triangleright$ The embedded query set
        $\mathcal{L} \leftarrow \frac{1}{|\mathcal{Q}^H|} \sum_{(h^*, y^*) \in \mathcal{Q}^H} -\log p(y^* \mid h^*, \mathcal{S}^H)$     $\triangleright$ Compute the episode's loss
        $\theta' \leftarrow \theta' - \epsilon \dfrac{\partial \mathcal{L}}{\partial \theta'}$        $\triangleright$ Update the model via gradient descent
        $\gamma' \leftarrow \gamma' - \epsilon \dfrac{\partial \mathcal{L}}{\partial \gamma'}$
        $\beta' \leftarrow \beta' - \epsilon \dfrac{\partial \mathcal{L}}{\partial \beta'}$
    **end while**

---

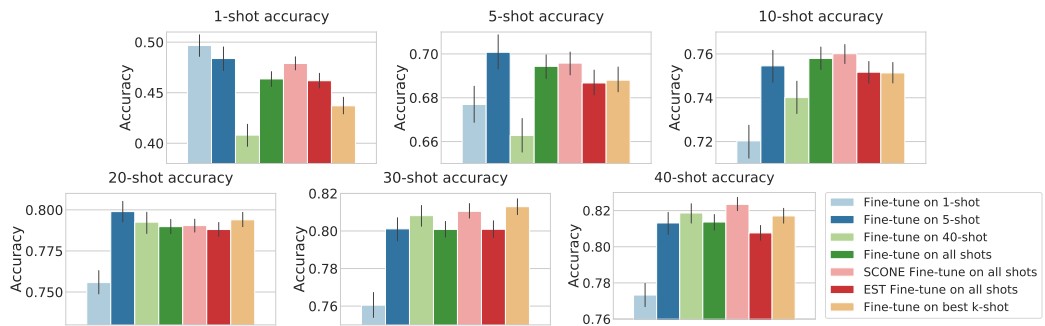

Figure 5: Additional evaluation shot settings to complement those in Figure 2 in the main paper. We refer the reader to Section 5.1 for a detailed description of these plots.

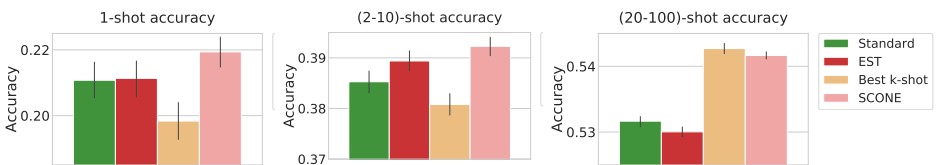

Figure 6: Break down of the performance in Table 1 in different evaluation shot ranges. We find that while the 'Best k-shot' baseline performs well for large shots (the third subplot), it performs poorly on low shots (first two subplots).

**Hypothesis testing**   We follow the same procedure as in (Triantafillou et al., 2020) to compute ranks for different methods that in turn determine which entries to bold in our tables. Specifically, we perform a 95% confidence interval statistical test on the difference between the mean accuracies of pairs of entries of each row. If for two entries we are not able to reject the null hypothesis that the difference between their means is 0, they will receive the same rank. For example, if model A and model B are tied for the first place according to that test, they will each receive the rank 1.5 (the average of the ranks 1 and 2). If we are able to reject that hypothesis, however, the entry with the larger mean accuracy will receive a higher rank than the other. In each row, we bold the entries that are tied for the highest rank. For convenience, in the last section of the Appendix, we show a more heavily-annotated copy of each table in the paper to make the rank computation procedure more transparent.

## ADDITIONAL RESULTS

First, we provide in Figure 5 additional plots to cover more evaluation shot settings than those shown in Figure 2 in the main paper. The setup for this is exactly the same as for Figure 2.

Next, since we observe that the 'Best k-shot' baseline performs well in Table 1, which reflects average performance across episodes of varying shots, we further break down its performance different ranges of shots in Figure 6. We find that while indeed 'Best k-shot' performs well for large shots, it actually performs poorly for low shots. This finding strengthens our case against this baseline: not only is it computationally expensive, requiring training multiple different models to pick the one that performs best, but is also is not as consistent as SCONE in its performance on different shot ranges.

Finally, to place our results into context, we display the results of Meta-Baseline with SCONE alongside the performance of recent work, controlling for model capacity, in Table 3. The approaches we compare against are: Classifier-Baseline  (Chen et al., 2020), SUR-pf (Dvornik et al., 2020), TaskNorm (Bronskill et al., 2020) and Simple CNAPs (Bateni et al., 2020). In particular, we report the performance of the parametric family of SUR ('SUR-pf') instead of full SUR (which has 8x more parameters), in order to make apples-to-apples comparisons with the remaining approaches. We find that the Meta-Baseline method, when combined with SCONE, achieves state-of-the-art on Meta-Dataset in this context, according to the average rank metric.

| Dataset | Classifier-Baseline | Meta-Baseline SCONE | SUR-pf | TaskNorm | Simple CNAPs |
|---|---|---|---|---|---|
| ILSVRC-2012 | $53.44 \pm 0.82\%$ | $53.69 \pm 0.83\%$ | $56.40 \pm 1.20\%$ | $50.60 \pm 1.10\%$ | $\mathbf{58.60 \pm 1.10\%}$ |
| Omniglot | $81.66 \pm 0.73\%$ | $90.01 \pm 0.49\%$ | $88.50 \pm 0.80\%$ | $90.70 \pm 0.60\%$ | $\mathbf{91.70 \pm 0.60\%}$ |
| Aircraft | $70.65 \pm 0.62\%$ | $78.27 \pm 0.54\%$ | $79.50 \pm 0.80\%$ | $\mathbf{83.80 \pm 0.60\%}$ | $82.40 \pm 0.70\%$ |
| Birds | $76.99 \pm 0.64\%$ | $\mathbf{79.62 \pm 0.58\%}$ | $76.40 \pm 0.90\%$ | $74.60 \pm 0.80\%$ | $74.90 \pm 0.80\%$ |
| DTD | $\mathbf{71.28 \pm 0.56\%}$ | $71.89 \pm 0.59\%$ | $\mathbf{73.10 \pm 0.70\%}$ | $62.10 \pm 0.70\%$ | $67.80 \pm 0.80\%$ |
| Quickdraw | $64.09 \pm 0.67\%$ | $71.95 \pm 0.56\%$ | $75.70 \pm 0.70\%$ | $74.80 \pm 0.70\%$ | $\mathbf{77.70 \pm 0.70\%}$ |
| Fungi | $50.23 \pm 0.81\%$ | $\mathbf{57.04 \pm 0.74\%}$ | $48.20 \pm 0.90\%$ | $48.70 \pm 1.00\%$ | $46.90 \pm 1.00\%$ |
| VGG Flower | $89.14 \pm 0.44\%$ | $\mathbf{91.09 \pm 0.39\%}$ | $\mathbf{90.60 \pm 0.50\%}$ | $89.60 \pm 0.60\%$ | $\mathbf{90.70 \pm 0.50\%}$ |
| Traffic Signs | $89.14 \pm 0.44\%$ | $70.33 \pm 0.56\%$ | $65.10 \pm 0.80\%$ | $67.00 \pm 0.70\%$ | $\mathbf{73.50 \pm 0.70\%}$ |
| MSCOCO | $\mathbf{53.92 \pm 0.78\%}$ | $52.94 \pm 0.82\%$ | $52.10 \pm 1.00\%$ | $43.40 \pm 1.00\%$ | $46.20 \pm 1.10\%$ |
| Average | 68.03% | 71.68% | 70.56% | 68.53% | 71.04% |
| Average | 3.45 | **2.40** | 2.85 | 3.75 | 2.55 |

Table 3: Comparison of our best SCONE model to recent state-of-the-art approaches on Meta-Dataset (for the setting of training on the training sets of all datasets).

## EXPERIMENTAL DETAILS

We plan to open source our code upon publication, including all experimental details. In the meantime, we outline these details below for completeness.

**Architecture**  We use ResNet-18 as the feature extractor for all of our experiments, following the implementation in (Triantafillou et al., 2020). For the SCONE variants, we add FiLM to all of the batch normalization layers throughout the network.

**Image processing**  For all experiments, we use Meta-Dataset's input pipeline to obtain images, and we follow the image processing performed in (Chen et al., 2020) which yields images of size 128 x 128. We apply standard data augmentation consisting of horizontal flipping and random cropping followed by standardization using a commonly-used mean and standard deviation as in (Chen et al., 2020). For episodic models, data augmentation is applied in both the support and query sets. No data augmentation is used at validation nor test time.

**Optimization**  We use ADAM with exponential learning rate decay and weight decay of $1e - 8$ to optimize all models in this work. We tune the initial learning rate, the decay rate, and the number of updates between each learning rate decay separately for each model presented in the paper. The initial learning rate values we considered are $0.0005$ and $0.001$, with a decay factor of $0.8$ or $0.9$ applied every $1000, 2000, 3000$ steps. We ran a variant for every combination of those values. We also tune the weight decay applied to the FiLM parameters (for SCONE variants) or the batch normalization parameters (for non-SCONE variants). We tried the values: $1e - 8$, $1e - 6$, $1e - 4$.

**SCONE hyperparameters**  For the SCONE variants, aside from the above hyperparameters, we additionally tune the smoothing parameter $m$ described in the main paper that is used for training and for evaluation. We did not tune the MAX-SHOT hyperparameter mentioned in the main paper as we found that our initial choices worked reasonably. Specifically, we set it to 40 for the smaller-scale experiments where the maximum shot was 40, and to 200 for the large-scale experiments. The latter choice was performed heuristically since shots much larger than 200 are unlikely under the shot distribution induced by Meta-Dataset's episode generator. For more information on that shot distribution, we refer the reader to the next section.

**SCONE smoothing hyperparameter**  We tuned the value of this hyperparameter that will be used both at training and at evaluation. At training time, we considered values in the range $0, 0.2, 0.4, 0.6, 0.9$ for Prototypical Network experiments, and we picked the variant that worked best according to the validation performance that was computed without smoothing. Once the model was trained and all of the remaining hyperparamters were tuned, we performed a final validation round to tune the evaluation-time smoothing that will be used in the chosen model. We found it beneficial to use larger values here, picking the value of $1 - 1e - 06$ for example for the Prototypical Network on ImageNet. In the Meta-Baseline codebase, we trained with larger values of smoothing (the best we found was $1 - 1e - 10$) and didn't find it beneficial to additionally smooth at evaluation time.

| | Prototypical Networks (ImageNet only) | | | | | |
|---|---|---|---|---|---|---|
| Dataset | Standard | L2 BN | EST | Best k-shot | SCONE w/o S | SCONE |
| ILSVRC-2012 | $50.90 \pm 1.12\%(6)$ | $\mathbf{51.81 \pm 1.06\%(3)}$ | $\mathbf{52.17 \pm 1.09\%(3)}$ | $52.36 \pm 1.08\%(3)$ | $\mathbf{52.98\% \pm 1.09\%(3)}$ | $52.51 \pm 1.11\%(3)$ |
| Omniglot | $63.12 \pm 1.37\%(5.5)$ | $63.14 \pm 1.32\%(5.5)$ | $\mathbf{66.07 \pm 1.29\%(2.5)}$ | $65.94 \pm 1.33\%(2.5)$ | $\mathbf{64.71\% \pm 1.32\%(2.5)}$ | $65.60 \pm 1.34\%(2.5)$ |
| Aircraft | $54.30 \pm 0.97\%(5.5)$ | $53.26 \pm 0.97\%(5.5)$ | $\mathbf{55.64 \pm 0.88\%(2.5)}$ | $56.03 \pm 0.95\%(2.5)$ | $\mathbf{55.38\% \pm 0.96\%(2.5)}$ | $55.38 \pm 0.96\%(2.5)$ |
| Birds | $68.22 \pm 0.97\%(5.5)$ | $\mathbf{69.21 \pm 1.01\%(2.5)}$ | $67.17 \pm 1.02\%(5.5)$ | $68.63 \pm 1.11\%(2.5)$ | $\mathbf{68.98\% \pm 1.04\%(2.5)}$ | $69.70 \pm 1.01\%(2.5)$ |
| DTD | $66.62 \pm 0.90\%(6)$ | $68.33 \pm 0.81\%(4.5)$ | $68.20 \pm 0.77\%(4.5)$ | $\mathbf{69.61 \pm 0.80\%(2)}$ | $\mathbf{68.68\% \pm 0.78\%(2)}$ | $69.58 \pm 0.77\%(2)$ |
| Quickdraw | $\mathbf{59.79 \pm 0.98\%(3)}$ | $59.17 \pm 0.96\%(6)$ | $\mathbf{60.05 \pm 0.97\%(3)}$ | $60.68 \pm 0.97\%(3)$ | $\mathbf{60.00\% \pm 1.00\%(3)}$ | $60.81 \pm 0.95\%(3)$ |
| Fungi | $36.77 \pm 1.07\%(5.5)$ | $\mathbf{38.96 \pm 1.10\%(2.5)}$ | $\mathbf{39.50 \pm 1.12\%(2.5)}$ | $37.96 \pm 1.08\%(5.5)$ | $\mathbf{39.19\% \pm 1.15\%(2.5)}$ | $39.66 \pm 1.12\%(2.5)$ |
| VGG Flower | $86.61 \pm 0.87\%(5)$ | $\mathbf{87.70 \pm 0.77\%(2)}$ | $\mathbf{88.55 \pm 0.65\%(2)}$ | $87.45 \pm 0.86\%(5)$ | $86.98\% \pm 0.80\%(5)$ | $\mathbf{88.03 \pm 0.73\%(2)}$ |
| Traffic Signs | $48.64 \pm 1.06\%(3.5)$ | $46.54 \pm 1.03\%(6)$ | $48.41 \pm 1.07\%(3.5)$ | $\mathbf{50.26 \pm 1.16\%(1)}$ | $47.61\% \pm 1.05\%(3.5)$ | $48.24 \pm 1.09\%(3.5)$ |
| MSCOCO | $\mathbf{43.02 \pm 1.09\%(3.5)}$ | $\mathbf{43.11 \pm 1.05\%(3.5)}$ | $\mathbf{43.45 \pm 1.05\%(3.5)}$ | $43.20 \pm 1.18\%(3.5)$ | $\mathbf{43.43\% \pm 1.08\%(3.5)}$ | $44.25 \pm 1.11\%(3.5)$ |
| Average | 57.80 % | 58.12 % | 58.92% | 59.21% | 58.79% | 59.38% |
| Average ranks | 4.90 | 4.10 | 3.25 | 3.05 | 3.00 | **2.70** |

Table 4: The same table as Table 1 additionally annotated with per-row ranks.

**Model selection** For each experiment, we perform early stopping according to the performance on the validation set. For the models that train on a single shot $k$ in the smaller-scale experiments, the validation performance that we monitor for early stopping is the average query set accuracy on $k$-shot 5-way episodes drawn from the validation set. For the models in the small-scale experiments that train on a distribution of shots, we use the average validation performance over 5-way episodes whose shot is sampled according to the same distribution used for training the respective model. For the larger-scale Meta-Dataset experiments, we draw validation episodes only from the validation set of ImageNet for the experiments that train on ImageNet only, or from the validation sets of all datasets for the experiments that train on all datasets. In both cases, the validation episodes are drawn using Meta-Dataset's episode generator that yields episodes of variable ways and variable shots with class imbalance. In all cases, the average validation performance is computed over 600 validation episodes and is monitored every 2K training updates. We apply exponential smoothing to the resulting validation "curve" (using the default value of 0.6 in TensorBoard). Then, we choose the update step at which the highest peak of that curve is found and we use the checkpoint corresponding to that update step for testing.

## DISTRIBUTION OF SHOTS IN META-DATASET EPISODES

For reference, Figure 7 displays histograms of the number of shots produced by Meta-Dataset's episode sampling algorithm. These are computed by sampling 600 episodes per dataset for each of the training, validation and test splits of Meta-Dataset.

## TABLES WITH MORE DETAILED RANKS

In this section, we include a copy of the same tables appearing previously in the paper, but additionally annotated with per-row ranks, to make the rank computation method more transparent. These tables are Table 4, Table 5 and Table 6.

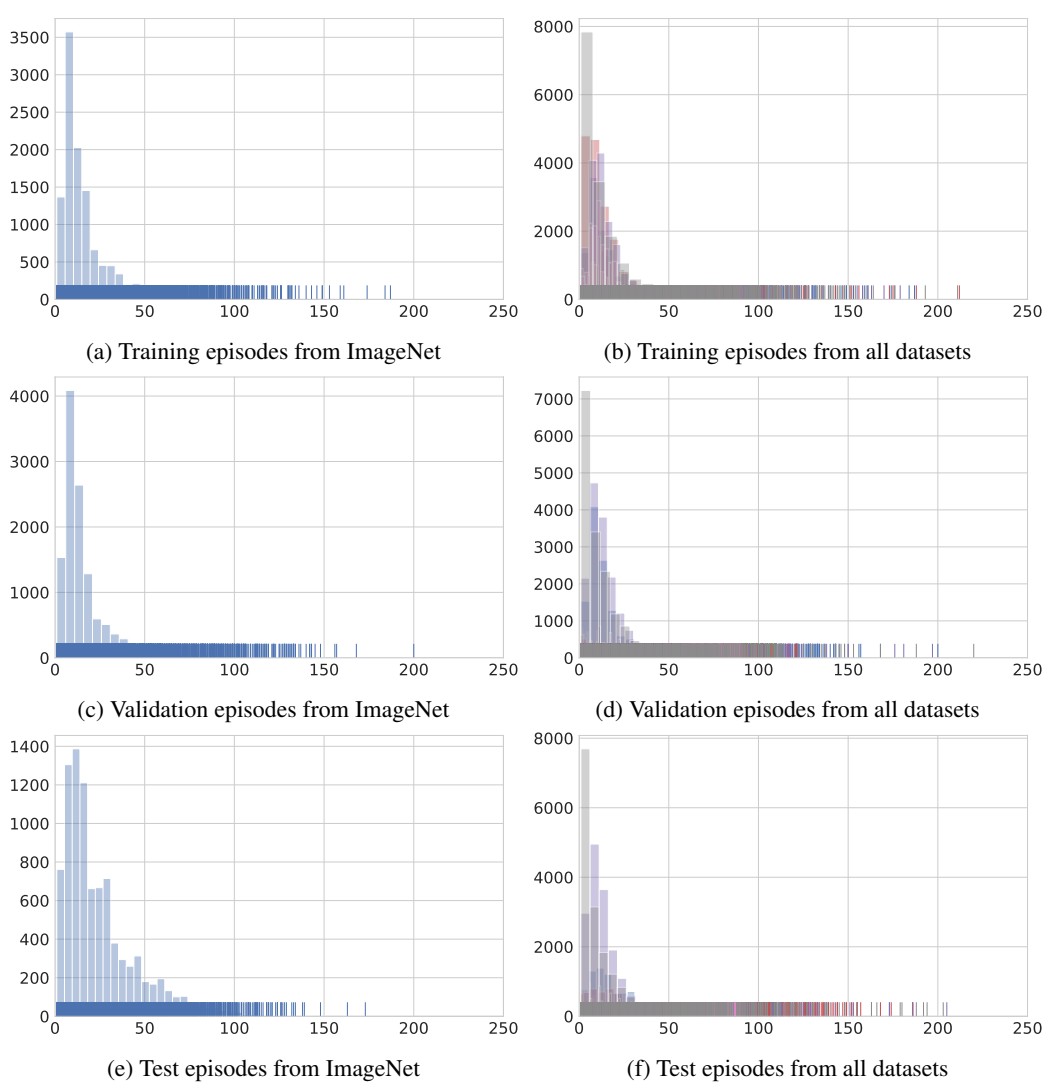

Figure 7: Histogram of shots appearing in episodes generated using Meta-Dataset's sampling algorithm for the different splits.

| | Meta-Baseline (All datasets) | | |
|---|---|---|---|
| Dataset | Classifier-Baseline | Control | SCONE |
| ILSVRC-2012 | **53.44 ± 0.82% (1.5)** | 49.83 ± 0.80% (3) | **53.69 ± 0.83% (1.5)** |
| Omniglot | 81.66 ± 0.73% (3) | **89.28 ± 0.51% (1.5)** | **90.01 ± 0.49% (1.5)** |
| Aircraft | 70.65 ± 0.62% (3) | **81.60 ± 0.49% (1)** | 78.27 ± 0.54% (2) |
| Birds | 76.99 ± 0.64% (3) | **78.75 ± 0.59% (1.5)** | **79.62 ± 0.58% (1.5)** |
| DTD | **71.28 ± 0.56% (1.5)** | 70.47 ± 0.58% (3) | **71.89 ± 0.59% (1.5)** |
| Quickdraw | 64.09 ± 0.67% (3) | **72.79 ± 0.59% (1.5)** | **71.95 ± 0.56% (1.5)** |
| Fungi | 50.23 ± 0.81% (3) | 55.28 ± 0.73% (2) | **57.04 ± 0.74% (1)** |
| VGG Flower | 89.14 ± 0.44% (3) | 90.13 ± 0.43% (2) | **91.09 ± 0.39% (1)** |
| Traffic Signs | 68.87 ± 0.61% (3) | **70.37 ± 0.56% (1.5)** | **70.33 ± 0.56% (1.5)** |
| MSCOCO | **53.92 ± 0.78% (1.5)** | 47.85 ± 0.81% (3) | **52.94 ± 0.82% (1.5)** |
| Average | 68.03% | 70.63% | 71.68% |
| Average rank | 2.55 | 2 | **1.45** |

Table 5: The same table as Table 2 additionally annotated with per-row ranks.

| Dataset | Classifier-Baseline | Meta-Baseline SCONE | SUR-pf | TaskNorm | Simple CNAPs |
|---|---|---|---|---|---|
| ILSVRC-2012 | $53.44 \pm 0.82\%$(3.5) | $53.69 \pm 0.83\%$(3.5) | $56.40 \pm 1.20\%$(2) | $50.60 \pm 1.10\%$(5) | $\mathbf{58.60 \pm 1.10\%(1)}$ |
| Omniglot | $81.66 \pm 0.73\%$(5) | $90.01 \pm 0.49\%$(2.5) | $88.50 \pm 0.80\%$(4) | $90.70 \pm 0.60\%$(2.5) | $\mathbf{91.70 \pm 0.60\%(1)}$ |
| Aircraft | $70.65 \pm 0.62\%$(5) | $78.27 \pm 0.54\%$(4) | $79.50 \pm 0.80\%$(3) | $\mathbf{83.80 \pm 0.60\%(1)}$ | $82.40 \pm 0.70\%$(2) |
| Birds | $76.99 \pm 0.64\%$(2.5) | $\mathbf{79.62 \pm 0.58\%(1)}$ | $76.40 \pm 0.90\%$(2.5) | $74.60 \pm 0.80\%$(4.5) | $74.90 \pm 0.80\%$(4.5) |
| DTD | $\mathbf{71.28 \pm 0.56\%(2.5)}$ | $71.89 \pm 0.59\%$(2.5) | $\mathbf{73.10 \pm 0.70\%(1)}$ | $62.10 \pm 0.70\%$(5) | $67.80 \pm 0.80\%$(4) |
| Quickdraw | $64.09 \pm 0.67\%$(5) | $71.95 \pm 0.56\%$(4) | $75.70 \pm 0.70\%$(2.5) | $74.80 \pm 0.70\%$(2.5) | $\mathbf{77.70 \pm 0.70\%(1)}$ |
| Fungi | $50.23 \pm 0.81\%$(2) | $\mathbf{57.04 \pm 0.74\%(1)}$ | $48.20 \pm 0.90\%$(3.5) | $48.70 \pm 1.00\%$(3.5) | $46.90 \pm 1.00\%$(5) |
| VGG Flower | $89.14 \pm 0.44\%$(4.5) | $\mathbf{91.09 \pm 0.39\%(2)}$ | $\mathbf{90.60 \pm 0.50\%(2)}$ | $89.60 \pm 0.60\%$(4.5) | $\mathbf{90.70 \pm 0.50\%(2)}$ |
| Traffic Signs | $89.14 \pm 0.44\%$(3) | $70.33 \pm 0.56\%$(2) | $65.10 \pm 0.80\%$(5) | $67.00 \pm 0.70\%$(4) | $\mathbf{73.50 \pm 0.70\%(1)}$ |
| MSCOCO | $\mathbf{53.92 \pm 0.78\%(1.5)}$ | $\mathbf{52.94 \pm 0.82\%(1.5)}$ | $52.10 \pm 1.00\%$(3) | $43.40 \pm 1.00\%$(5) | $46.20 \pm 1.10\%$(4) |
| Average | 68.03% | 71.68% | 70.56% | 68.53% | 71.04% |
| Average | 3.45 | **2.40** | 2.85 | 3.75 | 2.55 |

Table 6: The same table as Table 3 additionally annotated with per-row ranks.

