# OpenReview forum: "Learning Flexible Classifiers with Shot-CONditional Episodic (SCONE) Training"
_ICLR.cc/2021/Conference — Reject_

### Official Review · AnonReviewer4 · 2020-10-27
**“conditional on the number of shots” is somehow the main contribution which is however not significant, given no comparison to sota methods.**

**Rating:** 4
**Confidence:** 4

**Review:**

This paper proposed an implementation method of using different numbers of shots of data for few-shot learning such as to mitigate the negative effect of "different shots". It optimized the FiLM parameters using meta gradient descent during episodic meta-training with different-shot learning tasks. It conducted the experiments on quite a set of "meta-dataset benchmarks".

Pros:

1. The study of varying-shot task learning could be interesting, as the quantity of data in each task really matters for few-shot learning.

2. "Conditional on the shot number" is a reasonable idea.

Cons:

1. The motivation for conducting such a work is not new, and more importantly, the proposed method is too incremental compared to many existing works. (1) An existing work [Cao et al. ICLR’20] (also cited by this submission) studied the same problem both theoretically and empirically and proposed a similar linear transformation method to handle the varying-shot learning issue (Please kindly refer to section 3.3 in [Cao et al. ICLR’20]). This submission makes an incremental contribution of meta-learning transformation parameters conditional on shot numbers (using an existing method called FiLM). (2) Regardless of using such conditional, the idea of meta-learning feature transformation parameters (i.e. scaling and shifting parameters) has been well-deployed in existing few-shot learning papers [Sun et al. CVPR’19] (not cited in this submission) and [Oreshkin et al., 2018; Requeima et al., 2019; Bateni et al., 2019]. While as I can see from some other papers from the same authors of these related works, this transformation seems to be working well in many recent and similar tasks such as class-incremental learning (different numbers of samples for old or new classes) [Liu et al. CVPR’20] and semi-supervised few-shot learning tasks [Li et al. NeurIPS’19]. (3) Conditional on a single scalar ($k$ in this submission) is also not new in meta-learning, please kindly check [Shu et al. NeurIPS’19]. While I admit that this “conditional” is somehow the main contribution proposed in this submission for the few-shot settings. But I don’t think it is significant.

2. The experiments lack comparisons to the sota few-shot learning methods (in the same setting of using varying-shot tasks in episodic training). This is a very clear weakness of this paper. The paper does not prove that using the proposed conditional meta-training can achieve better performance of varying-shot learning over the sota few-shot learning methods. Table 1 makes me feel that this paper proposes a new BN method.

3. I want to talk more about varying shots. Actually, for different classes (different levels of hardness or diversity among the classes), the model may need different numbers of samples to learn a good representation for each of them (e.g. the common sense is that a model needs more samples for the non-rigid class "person" than the rigid class "spoon"). Given the high randomness of the class sampled in any meta-learning task, it is not clear how many shots a task really needs (for its all classes). So, I am a bit doubting "is it necessary to study the problems of varying-shot learning".

[Sun et al. CVPR’19] Meta-Transfer Learning for Few-Shot Learning

[Liu et al. CVPR’20] Mnemonics Training: Multi-Class Incremental Learning without Forgetting (details about feature transformation are given in the supplementary as the transformation method is not the contribution in this CVPR’20 paper)

[Li et al. NeurIPS’19] Learning to Self-Train for Semi-Supervised Few-Shot Classification (details about feature transformation are given in the supplementary as the transformation method is not the contribution in this NeurIPS’19 paper)

[Shu et al. NeurIPS’19] Meta-Weight-Net: Learning an Explicit Mapping For Sample Weighting (please check fig1(c) and its explanations)

---

> ### Author Response · Authors · 2020-11-13
> **Response to Reviewer 4**
>
> We thank Reviewer 4 for their feedback, and for pointing out missing references, which we added in our revised submission.
>
> ### Comparison to SOTA
>
> We have updated Figure 2 and Table 1 to include a comparison to Cao et al., whose EST approach also tackles the problem of shot robustness, but with a shot-invariant strategy. SCONE outperforms EST while also being more generally applicable (beyond Gaussian classifiers). We also added accuracies reported for other methods evaluating on Meta-Dataset in Table 3 where we show that our Meta-Baseline with SCONE model achieves SOTA on Meta-Dataset when controlling for model capacity (to make apples-to-apples comparisons).
>
> ### Our contribution is not incremental
>
> We strongly disagree with the assertion that our work is incremental. While our work bears resemblance with that of Cao et al. in terms of  the shared aim of shot robustness, we adopt a shot-conditional strategy (as opposed to Cao et al.’s shot invariant strategy). Moreover, our proposed approach can be applied to a wide range of meta-learners, which we show by applying it to Meta-Baseline (which uses a cosine nearest-centroid inference algorithm) and observing a significant improvement in performance. In contrast, Cao et al.’s theoretical results only apply to the Gaussian classifier case. Their proposed EST approach consists of a shot-invariant linear transformation of the embeddings, whereas SCONE performs a shot-conditional feature-wise affine transformation of the hidden representations, resulting in a shot-conditional non-linear transformation of the embeddings. The training objectives also differ significantly between the two approaches.
>
> We think dismissing the novelty of our work due to the superficial relationship to related work in terms of using FiLM as a conditioning mechanism is unfair. Our contribution lies elsewhere: we study the effect of episodic fine-tuning in terms of its ability to tune a model to specific shot settings, and we introduce a shot-conditional approach (which to our knowledge has not been explored for few-shot classification) to capitalize on our findings and train a model that doesn’t suffer from the trade-offs that we and Cao et al. identified and yields very strong performance on Meta-Dataset. FiLM simply happens to be the conditioning approach we chose due to its simplicity and effectiveness.
>
> Sun et al. learns a feature-wise transformation of the weights and biases (rather than the activations) to fine-tune a MAML initialization on an episodic objective. The scaling and shifting transformations learned serve the purpose of fine-tuning rather than conditioning on auxiliary information (such as the shot).
>
> While SCONE, TADAM (Orsehkin et al.), CNAPs (Requeima et al.), and Simple CNAPs (Bateni et al.) all train a task-conditional model, the task representation used for conditioning by TADAM, CNAPs, and Simple CNAPs discards the shot information by averaging over all support embeddings. In contrast, SCONE ignores the support embeddings (as we point out in our submission) and only looks at the shot distribution. In other words, SCONE conditions on completely different support set information, and constitutes a complementary approach to that of TADAM, CNAPs, and Simple CNAPs.
>
> Shu et al.’s Meta-Weight-Net conditions on scalar information like SCONE does, but the nature of that information (the loss at a certain time step), the part of the model that’s affected by conditioning (the loss being re-weighted), and the problem settings in which the model is applied (long-tailed class distributions, noisy labels) are different.
>
> We’ve added all suggested references to our revised paper.
>
> ### Different classes may need different shots
>
> This is a very interesting point, and perhaps future work can explore active learning based methods to choose the minimal necessary support set to learn each class at hand. However, the object of study in this work is the standard few-shot classification problem where at evaluation time the model does not get to choose how many shots it gets (this information is given in the support set). Studying the effect of the shot that was used at *training time*, however, is very important since as we show in this work (and is shown in Cao et al. too), training for a sub-optimal shot setting can lead to detrimental results in certain evaluation-time shot settings. Our main contribution is to offer an efficient and effective general model that doesn’t suffer from these trade-offs and yields strong results.

---

> > ### Comment · AnonReviewer4 · 2020-11-22
> > **Comparison to sota is still totally missing; Performance improvement over EST is too marginal.**
> >
> > First of all, thanks for the detailed feedback. It is helpful for me to better understand the paper. Unfortunately, some are still unconvincing for me.
> >
> > Comparison to SOTA
> >
> > Table 1 shows compared to Cao et al (EST method), this proposed one has a very marginal boost of 0.46%; if compared to the best k-shot, it is only 0.17%. But if comparing EST (or best k-shot) to the Standard, it has over 1% gain. This means the main gain is from varying-shot (which is from Cao et al and with a theoretical proof in that paper) but this proposed method of using conditional varying-shot is incremental and only based on some empirical results.
> >
> > The sota methods I mentioned are the top few-shot learning methods (e.g. [A]) which are trained in standard settings such as 1-shot, 5-shot and 10-shot, but typically without the pre-training on the whole ImageNet. A fair comparison to these sota methods can be in two ways: one using normal training and testing (transductive or inductive as long as consistency is holding in different methods), and the other one using varying-shot meta-training and then testing on the same few-shot tasks. Absolutely, for this comparison, authors should re-train a lot of sota models using the same initialization weights from ImageNet pre-training (note this pre-training is not commonly used in few-shot learning methods). In addition, it is needed to implement all methods on uniform architectures such as some of the popular ones --- ResNet12, ResNet18, and WRN-28-10. Btw, the prototypical net is a too old baseline to convince the few-shot learning communities. If the proposed method can make a significant boost on stronger baselines then it will be more impressive. While I don’t think this experimentation takes short time to conduct.
> >
> > [A] Shell Xu Hu, Pablo Moreno, Yang Xiao, Xi Shen, Guillaume Obozinski, Neil Lawrence, and Andreas Damianou. Empirical bayes transductive meta-learning with synthetic gradients. International Conference on Learning Representations, 2020

---

> > > ### Author Response · Authors · 2020-11-23
> > > **In strong disagreement with Reviewer 4's comment**
> > >
> > > We thank Reviewer 4 for their time and effort in responding to our rebuttal.
> > >
> > > We would really appreciate if Reviewer 4 would consider the following very important points:
> > >
> > > SOTA on mini-ImageNet versus Meta-Dataset, and why we use the latter
> > > --------------------------------------------------------------------------------------------------
> > >
> > > In this work, we use the Meta-Dataset benchmark because a) it is the most challenging and realistic benchmark for few-shot classification that we are aware of, and b) our goal is to study (and improve upon) models’ robustness to shot settings at test time, and Meta-Dataset includes a wide distribution of shots.
> > >
> > > Many of the methods claiming SOTA on mini-ImageNet are actually currently extremely challenging to scale to the size of a benchmark like Meta-Dataset. [A] is a good example of that: it is even more complex than MAML, which itself scales poorly, which is largely why SOTA methods on Meta-Dataset haven't been following this optimization-based meta-learning paradigm. It shouldn't fall on us to prove that these methods work on a large scale (and in fact it hasn't for all previous methods published on Meta-Dataset). In regular image classification, this would be like criticizing a paper that achieves SOTA on ImageNet for not comparing with a method proved to be SOTA on CIFAR-10 only. Small scale benchmarks like mini-ImageNet certainly have value in facilitating the experimentation with more complex ideas. However, they cannot serve to demonstrate that a method is a *scalable* solution to few-shot learning (in fact the 1-shot/5-shot learning setting has been criticized as being too contrived, for instance see this blog post: https://www.amazon.science/blog/iclr-the-ai-conference-that-helped-redefine-the-field), and that burden still lies on that method's authors, not on the rest of the community. Moreover, "normal training and testing", which presumably means 1-shot or 5-shot training/testing, doesn't make sense for our work, since we are targeting the more realistic setting where the shot varies at test time.
> > >
> > > The bottom line is this: we've taken a much more challenging few-shot learning benchmark (Meta-Dataset), and demonstrated the value of SCONE relative to the published methods for that benchmark, following the precedents established by all other methods that were run on Meta-Dataset.
> > >
> > > Nevertheless, we remark that SUR, which is one of the approaches we compare against in Table 3, is also among the top-performers on mini-ImageNet (see Table 3 in the Dvornik et al’s SUR paper versus the ResNet12 results in [A]).
> > >
> > > Incorrect statements made in Reviewer 4’s comment
> > > --------------------------------------------------------------------------------------------------
> > > RE: “This means that the main gain is from varying-shot (which is from Cao et al)...”. This statement is incorrect. Cao et al did not propose to vary the shot during training. Their EST method is applied on embeddings that were learned using fixed shots (please see Table 1 in Cao et al’s paper). This is one aspect in which our approach differs: we propose to train on a distribution of shots in order to learn a conditioning mechanism that can then on-the-fly modulate the embedding network appropriately for test episodes of a variety of shots.
> > >
> > > RE: “This pre-training is not commonly used in few-shot learning methods”. This is incorrect. Recent methods for few-shot classification often use pre-training. To name a few: SUR (Dvornik et al), CNAPs (Requeima et al), Simple CNAPs (Bateni et al), Meta-Baseline (Chen et al), [Qiao et al, CVPR 2018], [Rusu et al ICLR 2019] and the [Sun et al. CVPR’19] paper that Reviewer 4 pointed us to in their original review.
> > >
> > > Small gain compared to EST
> > > --------------------------------------------------------------------------------------------------
> > > Indeed SCONE has a small boost over EST in Table 1, but SCONE is a general approach, whereas EST’s theoretical grounding is limited to Gaussian classifiers. Please note that in addition to Table 1, we have applied SCONE to the Meta-Baseline model (Tables 2 and 3), obtaining very strong performance.
> > >
> > > RE: “The prototypical net is too old a baseline...”
> > > --------------------------------------------------------------------------------------------------
> > > We perform our initial experimentation using this model because of its simplicity. However, we also applied SCONE to the very recent Meta-Baseline model, which is one of the top-performers on Meta-Dataset (Tables 2 and 3).
> > >
> > > Additional references:
> > > ----------------------------
> > > Siyuan Qiao et al. Few-shot image recognition by predicting parameters from activations. CVPR 2018.
> > > Rusu et al. Meta-learning with latent embedding optimization. ICLR 2019.

---

> > > > ### Comment · AnonReviewer4 · 2020-11-23
> > > > **regarding above feedback**
> > > >
> > > > For "SOTA on mini-ImageNet versus Meta-Dataset, and why we use the latter"
> > > >
> > > > 1. If existing standard FSL methods (especially the shot-resiliency methods like EST (Cao et al. 2020) and LEO (Rusu et al. 2019)) perform good enough (i.e., can reach a close performance to this proposed method as shown in Table 1), what is the contribution of this submission, then?
> > > > 2. I rethink the FSL, and I want to say my understanding that FSL aims to learn a model that can fast adapt to target tasks (which are usually in a low data regime) given some basic knowledge of those tasks, e.g. the scale. So, I doubt the meaning of working on this so-called varying-shot/scale FSL. I understand that the methods more robust to different FSL settings must be better. But, is that really necessary to mix those settings within once training? OR is that really significant for real applications?
> > > >
> > > >
> > > > For "Incorrect statements made in Reviewer 4’s comment"
> > > >
> > > > 1. Sorry for this misreading about EST, however, EST itself is shot-resiliency (also mentioned in this submission). Table 1 shows that EST can achieve good performance in a fair setting while the proposed method indeed has a very limited margin over it. This is again my concern on the contribution of this submission.
> > > > 2. What I can read from this submission is that it used the whole ImageNet dataset (1,000 classes each with around 1,000 samples in training). However, the standard FSL works mostly used miniImageNet training tasks to build a pre-training dataset consisting of 64 classes each with 600 samples. So they are different things, and obviously, using the whole ImageNet gives a much better initialization which makes a lot of difference for model adaptation.
> > > >
> > > > For the other feedbacks, I gracefully disagree and so will insist on my original comments.
> > > >
> > > > Thanks!

---

### Official Review · AnonReviewer2 · 2020-10-28
**A simple method for well-motivated problem**

**Rating:** 6
**Confidence:** 3

**Review:**

Summary:
This paper proposes to use a shot-conditioned model that specializes in pre-trained few-shot learning model to a wide spectrum of shots. The proposed approach is simple but effective, it trains neural networks that conditioned on a support set of a different number of shots K during the episodic fine-tuning stage, with FiLM as the conditioning mechanism.


Pros:
1. This paper is clearly written, well-motivated, and thoroughly evaluated. It is an enjoyable reading experience .
2. The proposed model and training algorithm is simple but effective
3. Experiments are well designed, which first verifies that shot-conditioned few-shot learners can achieve relatively good performances on different Ks and then perform the large-scale evaluation.

Cons:
1. Smoothing the shot distribution section is overly simplified. It is very hard to understand algorithm 1 given the limited details presented in the method section. I would suggest explaining in detail the smooth-shot procedure in the paper.
2. Lack of ablation studies. There are two key designs of the few-shot learner presented in this paper, that does not have detailed ablation study results. The first is whether using the convex combination of FiLM parameters to obtain shot distribution s. The second is whether to use smoothing in shot distribution. How much are these two designs contribute to the model's final performances?
3. In figure 2, it seems that the method trained with 5-shot has very good performances in all three scenarios presented? Would it be possible that such a good K-shot can be always found and therefore we do not necessarily need this K-conditioned few-shot learner? For instance, just always train with 5-shot and evaluate different shots.
4. Instead of looking at the results of some instances of different K, it would be nice to have a comprehensive evaluation over different Ks. For instance, evaluating over all Ks from 1 to 40 and compute the average accuracy of each different model. To reduce the computation overhead, we can also try to evaluate Ks in {1, 10, 20, 30, 40}.
5. Comparison of state-of-the-art methods for meta-dataset experiments.
6. What is UMAP projection? The content of the experiments is not self-contained. How shall we read Figure 3?
7. Figure 4 is also not straightforward to understand. What is the value on the y-axis? What does it mean?


Minor:
1. Inconsistent formats for average performances in Table 1. It seems that the results of the first two columns is badly formatted?

---

> ### Author Response · Authors · 2020-11-13
> **Response to Reviewer 2**
>
> We thank Reviewer 2 for their thoughtful comments.
>
> ### Comparison to SOTA
>
> We have updated Figure 2 and Table 1 to include a comparison to Cao et al., whose EST approach also tackles the problem of shot robustness, but with a shot-invariant strategy. SCONE outperforms EST while also being more broadly applicable. We also added accuracies reported for other methods evaluating on Meta-Dataset in Table 3 where we show that our Meta-Baseline with SCONE model achieves SOTA on Meta-Dataset when controlling for model capacity (to make apples-to-apples comparisons).
>
> ### Finding a “good” K-shot instead of K-conditioned few-shot learner
>
> Reviewer 2 raises an interesting point about the fact that 5-shot seems to perform reasonably in certain plots of Figure 2. However, it’s not the best in the 30-shot and 40-shot evaluation plots and in general we don’t expect it to be the best for even larger shots (the 30-shot one is newly added in Figure 5 of the revised paper, as per the last recommendation of Reviewer 2). Generally, it is an interesting question whether a “good shot” can always be found such that training only on that shot suffices to perform well on evaluation tasks in a range of shots. We ran an experiment for this; please refer to our response in Reviewer 1, in the “Optimal k-shot” section. Importantly, though, even if an optimal training shot always exists, the process of discovering that shot is very computationally intensive as it requires training multiple models, each with a different training shot, and then picking the one that works best on average. We argue that using SCONE is preferable, since it alleviates the need to train multiple models and offers a solution that works well in general, without suffering disproportionately in any shot.
>
> ### More granular evaluation shot settings
>
> We have evaluated on shots {1, 10, 20, 30, 40} as suggested, and added these plots in Figure 5. These additional settings highlight the fact that the 5-shot model isn’t the best option for larger shot evaluation tasks (30- and 40-shot), which further supports the observation that trade-offs exist. We also included EST (Cao et al.) in these plots, as per the suggestion of other reviewers, which is a method that also aims at shot robustness These additional plots strengthen our claim that SCONE is able to perform well in general and comes close to the performance of the model that was trained exclusively for the nearest shot of each given evaluation shot.
>
> ### Smoothing ablation
>
> As per Reviewer 2’s suggestion, we included an ablation of our method without any smoothing of the shot distribution, in Table 1. We observe that this variant is still competitive, but not as strong as our full model.
>
> ### Clarifications
>
> We also thank Reviewer 2 for bringing up different points for clarification. We have revised the paper to include a more comprehensive explanation of the smoothing procedure in Section 3. We invite Reviewer 2 to please let us know if the explanation is still unclear and we would be very happy to elaborate further. We will also release our code upon publication.
>
> The convex combination of FiLM parameters is needed for episodes where different classes have different shots. The issue we are facing is the following: we ultimately need to pick a single set of FiLM parameters to use for each episode. When all classes appearing in the episode have the same shot k, then we simply pick the k’th set of FiLM parameters. However, when different classes have different shots, we need a heuristic to dictate how we should choose the FiLM parameters for the episode. Our choice is to use a convex combination of the FiLM parameters corresponding to the shots that appear in the episode, according to the frequency with which those shots appear. Therefore, this is not a component of our approach that we can remove, without replacing it with a different heuristic. If Reviewer 2 has another heuristic for this in mind, we would be happy to try it.
>
> UMAP is a recent low-dimensional projection technique similar to t-SNE which runs more efficiently and better preserves global structure. We refer Reviewer 2 to McInnes et al. (2018) for more details. Figure 3 takes the concatenation of all FiLM parameters associated with a given shot setting and projects it down to a point on the Cartesian plane, which allows us to visualize how different shot settings relate to each other in terms of their FiLM parameterization. Indeed we find that the parameters for different shots (represented by colours) are close to each other in the low-dimensional space, which serves as a useful sanity check.
>
> Finally, Figure 4 is illustrates the shot distribution after applying smoothing. The y-axis is the distribution mass for each of the shot settings (that are shown on the x-axis). This distribution is used to select (in a soft way) the FiLM parameters that will be used for the episode at hand. We would be happy to expand on this further if it is still unclear.

---

> > ### Comment · AnonReviewer2 · 2020-11-19
> > **Most of my concerns are addressed**
> >
> > Thanks to the authors for the detailed responses and clarification. The revised manuscript is more self-contained and has included more comprehensive experiments that study various ablations of the proposed method and compare more prior methods.
> >
> > I believe most of my concerns are addressed and would like to upgrade my score.

---

### Official Review · AnonReviewer1 · 2020-10-29
**Good experiments, but have conceptual issues and baseline comparisons needed**

**Rating:** 6
**Confidence:** 5

**Review:**

Summary

The paper proposes a solution to few-shot meta learning approaches overfitting to the number of shots they are finetuned on, and not generalizing as well as expected to novel shots. In order to mitigate this problem, the paper suggests a parameterization of the meta learner which also conditions on the number of shots the model trains on. In practice, this is done via manipulation of the batch normalization parameters based on the number of shots. With this conditioning, the paper shows that the models perform better across a range of shots that they are evaluated on, compared to various sensible baselines.

Strengths
+ The paper does tackle an important problem, a meta learner should not be overfitting to the number of shots
+ The paper has very thorough experiments and numerous visualizations (for e.g. of the learnt batch norm embeddings) to help gain better intuitions on what the method does
+ The paper is well written and easy to understand

Weaknesses

Important points for the rebuttal are marked with (*)

(*) The problem setting of changing the number of shots during meta-training is still somewhat artificial since one could still test the model on a different number of shots during the Query set evaluation. I wonder if in light of this issue, conditioning on the number of shots is getting at the root of the issue. One would rather want to have methods like Cao et.al. which aim to make the model invariant (or insensitive) to the number of shots as opposed to explicitly conditioning on them. Would be great if the authors could clarify this issue.

(*) Related to the above issue, a disadvantage of the proposed method over prototypical networks is that ProtoNets could be applied to episodes of any number of shots, even those unseen during training. Similarly, Cao et.al. can also be applied to any number of shots since it does a projection of the feature embedding. However the conditioning approach fails in this case, which is somewhat unrealistic. I understand that there is a smoothing step in the shot distribution which could partially help mitigate that issue, but it does not seem satisfying. How do the authors think this issue could be mitigated, and how should one view this as a potential shortcoming of the approach in light of the goals of the paper?

(*) In general, it seems appropriate for the paper to have a comparison to Cao et.al. who propose a solution to shot-overfitting in the context of prototypical networks. While I understand that the proposed approach is more general than that of Cao et.al. that is not sufficiently demonstrated via. the experiments which still focus on the prototypical networks, which is not really an issue, but then warrants comparison to Cao et.al.

(*) It would also be really useful to include a comparison to prototypical networks that does sum of the features instead of the mean to compute the prototype, which would preserve information on how many shots there exists in the support set. In general, it would also serve as a more stringent test of whether shot conditioning in the way the paper does it is needed when the method is able to deal with or recognize when the number of shots has changed, which the prototypical network approach is not able to do.

It would also help to guide discussion on how shot conditioning might look like for different algorithms. For prototypical networks we used FiLM, for something like MAML one could imagine unrolling for a different number of steps based on how many shots there are to prevent overfitting. A discussion on this would help improve the paper.

(*) It would be useful to compare against a single model which trains on “k-shots” and is evaluated on a mixture of k-shots (as in meta-dataset) and search explicitly for the best value of “k” from [1, MAX_SHOTS]. Does the proposed approach still outperform that “best” (over k) model? That seems to be an important baseline comparison.

** POST REBUTTAL UPDATE**

I went through the other reviews and the author rebuttals. I thank the authors for throughly addressing all of my concerns and
running some baseline experiments which I think are quite interesting, and add to the completeness of the paper, including a
comparison to EST and an optimal K-shot baseline. I think the idea is simple and interesting, and now the paper has enough
experimental comparisons to be a valuable addition to the conference. I have updated my score accordingly.

---

> ### Author Response · Authors · 2020-11-13
> **Response to Reviewer 1**
>
> We thank Reviewer 1 for their valuable feedback.
>
> ### Prototypical Networks with summation
>
> This is an interesting idea, as indeed averaging embeddings discards shot information. We implemented it and found that it yields very poor results, i.e. the query accuracy never improves when fine-tuning episodically from the supervised model checkpoint. We don’t have a definitive explanation, but we conjecture that it relates to the fact that distances between query embeddings and prototypes are now a function of the shot.
>
> ### “Optimal k”-shot
>
> We ran an experiment to find an “optimal k” training shot which performs well on average on a wide range of test shots. For this, we treated k as a hyperparameter and searched over values in {1, 5, 10, 20, 30, 40}. We have updated Figure 2 and Table 1 to include these results. Figure 2 shows that the best k-shot model still suffers from the observed trade-offs (the best k there was 15, so it performs poorly on 1-shot for instance). In Table 1, this baseline works well, though it still falls short of SCONE. Regardless, we emphasize that the advantage of using SCONE over this baseline is that it does not require an expensive hyperparameter search over many values of k; yet still performs well on a wide range of shots.
>
> ### Comparison to EST
>
> EST was originally evaluated on smaller-scale benchmarks (Omniglot, mini-ImageNet, tiered-ImageNet) using shot settings in the lower end of the spectrum (1-10), and its theoretical justification is limited to Gaussian classifiers (which incidentally highlights the general applicability of SCONE). We applied it to our Prototypical Networks models on our smaller-scale experiments and on Meta-Dataset. To achieve the best results, we had to extensively re-tune EST’s hyperparameter which controls the balance between the covariance of means and the mean of covariances, as we describe in Section 5.1. We show the results in Figure 2 and Table 1 of our updated submission. SCONE outperforms EST by a small margin while being applicable to more meta-learning approaches, such as the Meta-Baseline approach we consider in Table 2.
>
> ### Generalization to unseen shots
>
> Since we are free to sample episodes however we want during training, we design the shot distribution so that it covers as many shot settings as possible. If the shot is bounded, such as in Section 5.1, then this ensures that all shots will be seen during training. If the shot is unbounded, then all shots of value MAX-SHOT or larger are lumped into a single MAX-SHOT category. This ensures that inference is well-specified no matter which shot we encounter at test time. It does however introduce an implicit assumption that shots of the MAX-SHOT magnitude or larger require the same processing in the embedding function. If that assumption does not hold in practice, then SCONE could benefit from more sophisticated strategies that attempt to extrapolate FiLM parameter values to larger shot settings; we think this would constitute an interesting avenue for future work.
>
> ### Shot-invariance versus shot-awareness
>
> Reviewer 1 brings up a very interesting point regarding the dilemma of shot-invariance (as in Cao et al.’s EST), versus shot-awareness (as in SCONE), which are two different strategies for building shot resiliency. A similar dilemma comes up in other areas too, e.g. in algorithmic fairness one can attempt to be invariant to sensitive attributes (in order to avoid being influenced by them), or one can attempt to become aware of them as a means of ensuring fair treatment with respect to those attributes. Ultimately, we believe that both avenues are valid and interesting and it is an empirical question which approach is more effective. Both strategies have potential weak points: Invariant strategies assume that the existence of a model that performs well while still being invariant to the variable of interest (in this case, the shot setting). If this assumption does not hold, this strategy may not be very effective. Conditional strategies, on the other hand, assume that the distribution of tasks seen during training (in this case, the distribution of shot settings) is representative of the test distribution, such that the conditional model can generalize to new values of the conditioning variable. If the test distribution varies too much, then a conditional strategy may break down. Our empirical results support the effectiveness of SCONE, but these questions are very interesting and would be happy to discuss more.
>
> ### How to apply SCONE to other meta-learners
>
> We agree that making other meta-learning algorithms shot-aware is interesting. SCONE is actually algorithm-agnostic, so it could be applied to various meta-learners, but indeed perhaps algorithm-specific adjustments (like the one Reviewer 1 proposes for MAML) may yield additional gains over SCONE’s  architectural modification of FiLM conditioning. We have added this point to our revised Conclusion section as an avenue for future work.

---

### Official Review · AnonReviewer3 · 2020-10-29
**Not familiar with this area, but assigned to review**

**Rating:** 5
**Confidence:** 1

**Review:**

This paper aims to understand the role of this episodic fine-tuning phase and discovers that fine-tuning on episodes of a particular shot can specialize the pre-trained model to solving episodes of that shot at the expense of performance on other shots. It proposes a shot-conditional form of episodic fine-tuning.

The main contribution of this work is the training objective, which varying the shots by introducing a distribution over shots $P_k$. In addition, the model parameters are not separately or independently maintained for different shots. The author proposed a conditioning mechanism using FiLM, where $k$ is a conditional variable or input to the FiLM network ($\gamma$ and $\beta$). The idea of shot-specific feature extractors is not new, and it is a common trick in amortized variational inference to reduce the number of model parameters.

In the experimental section, I did not see other baselines in previous works. The comparisons are mostly against the two methods named standard and L2 BN, which seem to be simple variation of Prototypical Networks. Even the proposed SCONE is a variation of Prototypical Networks, it also reduces the novelty or originality.

I am not familiar with this area or related research papers, but was assigned to review this paper. My evaluation is only based on the my understanding of the contents including in the manuscript. I would like to see the comments from other reviewers in this domain.

---

> ### Author Response · Authors · 2020-11-13
> **Response to Reviewer 3**
>
> We thank Reviewer 3 for their feedback. We hope our response addresses the three main points raised in their review.
>
> ### Contribution
>
> Our main contribution is to investigate the role of episodic fine-tuning from the perspective of its ability to tune a model to a specific shot setting. We both show the existence of this effect empirically and propose a way to leverage it to improve existing approaches. This investigation is timely and beneficial, because understanding the role of episodic fine-tuning is important in making forward progress in the context of many recent works which question the value of episodic training and other recent works which find value in applying a combination of supervised training followed by episodic fine-tuning.
>
> ### Novelty
>
> Can Reviewer 3 clarify the connection to amortized variational inference? What would be the “shot” analogue in this context, and does Reviewer 3 have specific papers in mind? In any case, we note that to the best of our knowledge, shot-conditional approaches have not been explored before in the context of few-shot classification.
>
> As Reviewer 1 pointed out, SCONE can be applied to meta-learners beyond Prototypical Networks, as shown by applying it on Meta-Baseline, which uses a nearest-centroid formulation with a cosine distance metric.
>
> ### Baselines and SOTA
>
> We have updated Figure 2 and Table 1 to include a comparison to Cao et al., whose EST approach also tackles the problem of shot robustness, but with a shot-invariant strategy. SCONE outperforms EST while being more general. We also added accuracies reported for other methods evaluating on Meta-Dataset in Table 3 where we show that our Meta-Baseline with SCONE model achieves SOTA on Meta-Dataset when controlling for model capacity (to make apples-to-apples comparisons).

---

### Author Response · Authors · 2020-11-19
**We would love to hear all reviewers' thoughts on our latest updates**

To all reviewers: we would love to hear your thoughts on our responses, especially the new baselines and ablations we ran, comparison with the Cao et al’s EST model and recent approaches on Meta-Dataset where we show that we achieve state-of-the-art when controlling for model capacity.

We also have a new update on our latest results: while the Best k-shot baseline performs well in Table 1 which reflects the average performance, we break down its performance in different ranges of test-time shots and show that, unlike SCONE, this baseline performs poorly for low shots. This strengthens our claims that SCONE is preferable: it not only alleviates the need to train multiple models (we only train once!) but performs consistently well on different shot ranges. We show the results of this new analysis in Figure 6 of our latest revision.

To Reviewer 3: We would love to know whether we addressed Reviewer 3’s concerns, especially around the novelty of our contribution, and comparison to relevant baselines and recent methods in this area.

To Reviewer 1: We ran the suggested baselines and comparisons (e.g. the “Best k-shot” baseline, and comparison with EST) and discussed the points raised in Reviewer 1’s review. We would love to hear back on whether their concerns are addressed satisfactorily.

To Reviewer 2: We have added a comparison to recent methods, analysis and experimentation on finding a “good k-shot”, more granular evaluation results, a smoothing ablation, and a number of clarifications. We would love to know if we’ve addressed Reviewer 2’s concerns adequately.

To Reviewer 4: We have discussed the difference between our work and the papers brought up, and added those to our revised paper. Our comparisons to recent work show we achieve SOTA when controlling for model capacity. We would love to hear from Reviewer 4 on whether we have addressed their concerns and to continue the discussion if not.

---

### Decision · Program_Chairs · 2021-01-07
**Final Decision**

**Decision:**

Reject

**Comment:**

The paper proposed a shot-conditional form of episodic fine-tuning approach for few-shot image classification. There were a number of concerns raised, e.g., there lacks of sufficient comparison with SOTA baselines, the justification on the significance of shot-aware approach is not entirely convincing, and incremental contributions in both novelty and improvements. While some of these issues were improved in the rebuttal, the revision remains not satisfied by the reviewers. Overall, I think the paper has some interesting idea, but is still not ready for publication.